# The Potential Utility of Circulating Oncofetal H19 Derived miR-675 Expression versus Tissue lncRNA-H19 Expression in Diagnosis and Prognosis of HCC in Egyptian Patients

**DOI:** 10.3390/biom13010003

**Published:** 2022-12-20

**Authors:** Shimaa Abdelsattar, Dina Sweed, Hala F. M. Kamel, Zeinab A. Kasemy, Abdallah M. Gameel, Hassan Elzohry, Omnia Ameen, Eman Ibrahim Elgizawy, Ahmed Sallam, Asmaa Mosbeh, Mahmoud S. Abdallah, Fatma O. Khalil, Hiba S. Al-Amodi, Sally M. El-Hefnway

**Affiliations:** 1Clinical Biochemistry and Molecular Diagnostics Department, National Liver Institute, Menofia University, Shebin El-Kom 32511, Egypt; 2Pathology Department, National Liver Institute, Menofia University, Shebin El-Kom 32511, Egypt; 3Department of Biochemistry, Faculty of Medicine, Umm Al Qura University, Makka 21955, Saudi Arabia; 4Department of Medical Biochemistry and Molecular Biology, Faculty of Medicine, Ain Shams University, Cairo 11566, Egypt; 5Public Health and Community Medicine Department, Faculty of Medicine, Menofia University, Shebin El-Kom 32511, Egypt; 6Clinical Pathology Department, National Cancer Institute, Cairo University, Cairo 11796, Egypt; 7Hepatology and Gastroenterology Department, National Liver Institute, Menofia University, Shebin El-Kom 32511, Egypt; 8Physiology Department, Faculty of Medicine, Menofia University, Shebin El-Kom 32511, Egypt; 9Department of Hepatobiliary and Pancreatic Surgery, National Liver Institute, Menofia University, Shebin El-Kom 32511, Egypt; 10Clinical Pharmacy Department, Faculty of Pharmacy, University of Sadat City (USC), Sadat City 32897, Egypt; 11Clinical and Molecular Microbiology and Immunology Department, National Liver Institute, Menoufia University, Shebin El-Kom 32511, Egypt; 12Department of Medical Biochemistry and Molecular Biology, Faculty of Medicine, Menofia University, Shebin El-Kom 32511, Egypt

**Keywords:** hepatocellular carcinoma (HCC), long noncoding RNA-H19 (lnRNAs-H19), microRNA (miR-675), real-time quantitative reverse transcription PCR (RT-qPCR)

## Abstract

Hepatocellular carcinoma (HCC) is one of the most common cancers worldwide. Interestingly, lncRNA-H19 acts independently in HCC and influences miR-675 expressions. We aimed to assess the potential utility of tissue lncRNA-H19 versus miR-675 expressions as a non-invasive biomarker for HCC diagnosis and prognosis in Egyptian patients. Ninety-one HCC patients and 91 controls included in this study were investigated for expression of lncRNA-H19 and miR675 using RT-qPCR. Our results showed that the expression of lncRNA-H19 and microRNA-675 were higher in patients than in controls (*p* < 0.001 for both). Additionally, lncRNA-H19 expression was higher in tumorous than in non-tumorous tissue (*p* < 0.001). Linear regression revealed that miR-675 expression was a significantly higher positive predictor than lncRNA-H19 for tumor size, pathologic grade, and AFP level; similarly, for cyclin D1 and VEGF protein expression. By using the ROC curve, the sensitivity of miR-675 was higher than lncRNA-H19 for discriminating HCC from controls (95–89%, respectively) and the sensitivity of lncRNA-H19 was higher in tumorous than in non-tumorous tissues (76%). The high expressions of both were associated with low OS (*p* < 0.001, 0.001, respectively). Oncofetal H19-derived miR-675 expression could be considered a potential noninvasive diagnostic and prognostic biomarker, outstanding the performance of the expression of tissue lncRNA-H19 for HCC.

## 1. Introduction

Hepatocellular carcinoma (HCC) is the most prevalent form of liver cancer and could be considered one of the main reasons for cancer-related deaths globally [1,2]. HCC appears to be caused partly by genetic instability or mutations. These could disrupt signaling pathways and affect healthy liver cells, which mainly cause liver tissue damage. Cirrhosis patients are more likely to develop HCC [3].

Overall, HCC patients have a poor prognosis [4], mainly due to inadequate treatment and delayed diagnosis [5,6]. However, the vast majority of HCC screening techniques, such as ultrasound and serum alpha-fetoprotein (AFP), are insufficiently sensitive and specific [7], so they are not comprised within clinical practice guidelines [8]. Hence, there is a need to identify biomarkers that can predict HCC or enable early diagnosis [1].

Long noncoding RNAs (lncRNAs) have gained attention in molecular biology with the development of sequencing technologies for the whole genome and transcriptome. One of the foremost lncRNAs to be identified was lncRNA-H19; it is highly expressed in organs of the developing body and is absent or weakly expressed in most adult tissues [9]. There is growing evidence that lncRNA-H19 is essential for embryogenesis [10]. Although several studies reported that lncRNA-H19 levels are elevated in several malignancies such as breast, brain, hepatic, lung, gastric, prostate, and urinary bladder cancers, there have been contradicting data on whether lncRNA-H19 promotes tumor growth or suppresses it [11,12,13,14]. According to most studies, tumor growth, migration, invasion, and/or metastasis have been linked to lncRNA-H19; however, the reported functional mechanisms of lncRNA-H19 vary depending on the type of cancer. Furthermore, it has been proposed that lncRNA-H19 expression levels in patients with specific cancers may aid in diagnosis and prognosis [15].

The primary precursor of microRNA (miR-675) is lncRNA-H19, which is embedded within the first exon of the H19 gene. The miR-675 is negatively regulated by the mRNA-binding protein ELAV like RNA-binding protein 1 (ELAVL1/HuR), which has been involved in inflammatory and carcinogenesis processes [16]. H19 is found to act as an independent lncRNA, additionally, it influences the levels of this miR-675 to some extent [17]. LncRNA-H19 can regulate various biological processes via the H19/miR-675 axis by targeting oncogenic or tumor suppressive factors because miR-675 has numerous targets in various signaling pathways [18,19].

It is interesting to note that a growing body of research has shown that lncRNAs function as a novel class of regulatory molecules involved in the initiation and development of hepatocellular carcinoma. For example, LncRNA H19 impacts the cell cycle, invasion, tumorigenicity, migration, apoptosis, and metastasis [20]. Additionally, they have been linked to altering vascular function and angiogenesis [21].

Hypervascularization, which highlights that HCC produces various angiogenic elements with a high tendency to invade the vasculature, is a notable characteristic of most HCC. Vascular endothelial growth factor (VEGF), and Epidermal growth factor receptor (EGFR), which act on both tumor and endothelial cell populations, are two of the growth factors that control the angiogenic response in HCC [22]. Cyclin D1 and CDK-activating kinase (CAK) trimeric are associated with cell-cycle progression in HCC [23]. They are all angiogenic markers and could all be used as HCC therapeutic targets [24].

Mounting evidence indicates that the diagnosis of hepatocellular carcinoma may be impacted by the aberrant expressions of lncRNAs [25]. However, most studies have concentrated on the function of lncRNA-H19 in cancer. Little research has been done on miR-675 [26,27]. Therefore, the current study aimed to assess the potential utility of tissue lncRNA-H19 expression and circulating miR-675 as a non-invasive biomarker for potential HCC diagnosis and prognosis, as well as assessing changes in EGFR, VEGF, and cyclin D1 levels in hepatic tissue.

## 2. Materials and Methods

In this case-control study, ninety-one patients with previously diagnosed HCC were included. They underwent partial hepatectomy at the Hepatobiliary and Pancreatic Surgery Department, National Liver Institute, Menofia University, Egypt during which liver tissue samples were taken. A control group of 91 healthy potential donors was also recruited from the liver transplant unit. The control group had normal liver function tests, normal liver and biliary system ultrasounds, negative serological results for autoimmune and viral liver diseases, and no history of diabetes. The research was carried out between December 2019 and October 2021. The exclusion criteria for the HCC group included: Patients with a history of liver transplantation or those who have previously undergone treatment for HCC or exhibit thyroid, diabetes, or renal insufficiency. All patients and controls provided their written informed consent after approval of the study protocol by the National Liver Institute, Menoufia University ethics committee (NLIIRB protocol Number 00226/2020).

### 2.1. Specimen Collection

Blood samples were collected before any surgical intervention; using disposable syringes and sterile venipuncture, 10 mL of venous blood samples were collected. The blood samples were delivered in a plain test tube with a vacutainer. After allowing enough time for the blood to clot, 6 mL of serum was separated, and samples were divided into two fractions: 4 mL for liver function tests, and the remaining 2 mL was stored at −80 °C until it was time to assess miR-675 expression in serum.

Liver tissue samples were taken from 91 HCC cases and categorized into two subgroups: tumorous and non-tumorous (from the corresponding non-tumor liver tissue) approved by the histopathological assessment. In addition, 91 samples of healthy liver tissue were taken from healthy potential donors for liver transplantation. Fresh samples were taken from the tissue biorepository at the institution. As part of the usual clinical care for the patients, the corresponding formalin-fixed, paraffin-embedded (FFPE) specimens were obtained from the pathology department’s archive.

### 2.2. Assessment of miR-675 Expression by RT-qPCR

Real-time quantitative reverse transcription polymerase chain reaction (RT-qPCR) was used to assess the expression of circulating miRNA-675. A miRNeasy extraction kit (QIAGEN, Germantown, MD, USA, cat no. 217061) was used according to the manufacturer’s instructions to extract total RNA, including miRNA, from serum samples. The extracted RNAs were checked for purity and concentration using the NanoDrop 2000 spectrophotometer device from Thermo Scientific (USA), and then the extracted RNA was stored at −80 °C. A miScript II RT kit (QIAGEN, USA, cat no. 218161) was used according to the manufacturer’s instructions to create complementary DNA (cDNA) from extracted RNA (10 μL). Each reaction took place in a net volume of 20 μL on ice, and the following ingredients were pipetted into each well: 4 μL of miScript HiSpec RT buffer, 2 μL of miScript Nucleics Mix, 2 μL of miScriptTM reverse transcriptase, 2 μL of nuclease-free water. This was followed by 10 μL of extracted microRNA. Finally, the reverse transcriptase was inactivated for one cycle by incubating at 37 °C for 60 min and 95 °C for 5 min in a 2720 Applied Bio-systems thermal cycler (Singapore). RT-qPCR was performed for the detection of miRNA-675 as well as SNORD (reference gene) using miScript SYBR Green PCR kit (QIAGEN, USA, cat no. 218073). The total volume of the reaction was 25 μL: 12.5 μL of SYBR Green Master Mix, 3.5 μL of nuclease-free water, 4 μL of cDNA, 2.5 μL of miScript universal primer, and 2.5 μL of miScript primer assay were used. Mature miRNA-675 and mature SNORD (miScript primer assay kit, QIAGEN, USA) were amplified by using the following cycling conditions with the ABI7500 real-time PCR instrument (Applied Biosystems, Thermo Fisher scientific, Waltham, MA (Massachusetts), USA) software 2.0.1: an initial activation step at 95 °C for 15 min, followed by three steps of 40 cycles at 94 °C for 15 s, 55 °C for 30 s, and 70 °C for 30 s. Using the 2^−ΔΔCt^ method, the expression levels of miRNA-675 were normalized to those of SNORD.

### 2.3. Assessment of lncRNA H19 Expression by RT-qPCR

RT-qPCR was used to assess the expression of lncRNA-H19 in liver tissue quantitatively. First, the samples (tissue) were prepared. Then, the RNeasy plus Universal Kit (Qiagen, Germantown, MD, USA, cat no. 73404) was used according to the manufacturer’s instructions to extract total RNA from the tissue. Next, the quality and purity of the RNA were ensured using a Nandrop 2000 spectrophotometer device from Thermo Fisher Scientific (Waltham, MA (Massachusetts), USA). RNA was stored at −80 °C until it was needed. Then, the reverse transcription step was performed using a QuantiTect Reverse Transcription Kit (Qiagen, USA, cat no 205311) according to the manufacturer’s instructions. Next, RT-qPCR was used for the detection of miRNA-675 as well as GAPDH (reference gene) by using SensiFASTTM SYBR Lo-ROX Kit (Bioline, Meridian, Bioscience, Cincinnate, OH, USA, cat no. Bio-74001). Finally, relative quantification (RQ) of lncRNA-H19 gene expression was performed using the following designed primers (Midland, TX, USA): lncRNA-H19 forward primer: 5′TCCCAGAACCCACAACATGA 3′ and lncRNA-H19 reverse primer, 5′-TTCACCTTCCAGAGCCGATT 3′. GAPDH Forward primer: 5′GAAGGTGAAGGTCGGAGTC3′. Reverse primer: 5′GAAGATGGTGATGGGATTTC 3′. The total volume of the reaction for lncRNA-H19 and GAPDH was 20 μL consisting of 10 μL of SYBR green Master Mix, 1 μL of Nuclease-free water, and 6 μL of template cDNA in addition to 1.5 μL of each primer (forward and reverse).

Tissue lncRNA-H19 and the reference gene GAPDH were amplified by using the following cycling conditions with the ABI7500 real-time PCR instrument (Applied Biosystems) software 2.0.1: 95 °C for 5 min as an initial activation phase followed by 45 cycles at 95 °C for 20 s; 60 °C for 30 s; 72 °C for 1 min; then a final extension phase at 72 °C for 10 min. Relative quantitation (RQ) of the lncRNA-H19 gene was carried out by normalizing the amount of the target lncRNA-H19 gene to an endogenous reference gene (GAPDH) and relative to the control by the 2^−ΔΔCT^ method.

### 2.4. Pathological Studies

The 8th edition of the American Joint Committee on Cancer Stage Systems and the 5th edition of the WHO classifications of digestive tumors were used for the histopathological evaluation of HCC cases [28,29].

### 2.5. Assessment of VEGF. EGFR and Cyclin D1 by IHC

Immunohistochemical staining was used to assess the expression of epidermal growth factor receptor (EGFR), vascular endothelial growth factor (VEGF), and cycling D1 levels in tumorous HCC tissue, non-tumorous tissue, and normal tissue from the healthy donors. Each tumor had at least two representative cores, and each core also contained one core of non-tumor tissue that matched the tumor [30]. Sections from the complete samples were used for the normal liver tissue. A rabbit polyclonal VEGF antibody (1:200), a cyclin D1 antibody (1:200), and an EGFR antibody (1:200) were all obtained from (Bioss, Woburn, MA, USA). Tris-EDTA antigen retrieval with high PH (Dako, Ref K8000, Glostrup, Denmark) was applied to the 4-micron thick tissue section, which was mounted on a slide for 20 min at room temperature. The primary antibodies were incubated on the slides for an entire night at 4 °C. The 3-diaminobenzidine (DAKO) chromogen was used to detect the immunostaining using the EnVision FLEX/HRP detection system (DAKO A/S, Glostrup, Denmark). Positive and negative controls were applied for each run of the immunohistochemistry protocol to ensure its effectiveness. Cyclin D1 was located in the nucleus of hepatocytes, EGFR in the cytoplasm of hepatocytes, and VEGF in the endothelial cells. The Histoscore (H score) system produces a final score between 0 and 300 by multiplying the staining intensity (0–3) by the percentage of stained cells.

### 2.6. Sample Size

Concato et al. [31] and Peduzzi et al. [32] reported that the idea of an event per variable (EPV) of 10 is appropriate for Cox regression. This is because only independent variables with significant effect sizes are used in Cox regression [3,4]. Therefore, if the effect size is between medium and large, a lower rule of thumb, like EPV of 10, is still applicable.

### 2.7. Statistical Analysis

SPSS version 28.0 [SPSS Inc., Chicago, IL, USA] was used for the analyses. The Shapiro–Wilk test was carried out to determine the normality of the distribution. The Jonckheere–Terpstra test was used in linear trend analysis to determine whether there was a rising or falling trend among the ordered groups. By estimating the effect size following the Jonckheere–Terpstra [J-T] Test, the Mann–Kendall [M-K] test is used to identify the presence of linear or non-linear trends [steadily increasing/decreasing or unchanging] in a series of data. The sensitivity, specificity, accuracy, positive predictive value, and negative predictive value are all estimated using ROC curve analysis. Path diagrams were used in conjunction with linear regression analysis. The strength and direction of the relationship between the examined markers were evaluated using Spearman correlation. A Kaplan–Meier curve was shown to find the independent predictors for low survival, followed by a Cox regression analysis. *p*-values < 0.05 are considered statistically significant.

## 3. Results

Ninety-one Egyptian patients with HCC and 91 healthy controls were included in this study.

### 3.1. The Clinical and Pathological Features of the Studied HCC Patients

The clinical and pathological features of the studied HCC patients are demon-strated in Table 1. They were 69 males and 22 females; the mean age was (58.62 ± 5.84).

### 3.2. Biochemical and Histopathological Markers among the Studied Groups

Tissue lncRNA-H19 and circulating miR-675 were measured by RQ (2^−ΔΔCT^) in all patients and the control group (Figure 1). In circulating miR-675, the RQ value in patients ranged between 1.5 and 18.3 with a median RQ value of 11.6; in the control group, the RQ value ranged between 0.9 and 1.1, and the median RQ value was 1. The expression of Circulating miR-675 was significantly higher in HCC patients than control in the control group (*p* < 0.001). In tissue lncRNA-H19, the RQ value in the tumorous tissue of patients ranged between 10.3 and 33.5 with a median RQ value of 19.5, while in the non-tumorous tissue of patients, the RQ value ranged between 1.9 and 10.9, and the median RQ value was 4.5. In the control group, the RQ value ranged between 0.8 and 1.2, and the median RQ value was 0.9. The expression of tissue lncRNA-H19 was significantly high within HCC tumorous tissues than that of non-tumorous and the control group (*p* < 0.001). Both tissue lncRNA and circulating miR-675 were positively correlated with each other (Table 2). Serum AFP was significantly higher in HCC patients than in the control group (*p* < 0.001). EGFR, cyclin D1, and VEGF expression were significantly higher in the tissues of the tumorous group as compared to the adjacent non-tumorous liver and the control groups (*p* < 0.001, for all) (Figure 2).

### 3.3. The Correlation between lncRNA-H19 and miR-675 with Demographic and Biochemical Markers among the HCC Patients

A Spearman correlation analysis revealed that over-expression of tissue lncRNA-H19 and circulating miR-675 were significantly associated with old age, serum AFP level, large tumor size, and poor pathological grade. In addition, there was a positive correlation between them and the expression of EGFR, cyclin D1, and VEGF. Additionally, tissue lncRNA-H19 and circulating miR-675 significantly correlated (Table 3).

### 3.4. Path Analysis for Tumorous Samples

Linear regression analyses using a path diagram revealed that circulating miR-675 was a significantly higher positive predictor than tissue lncRNA-H19 to tumor size, pathologic grade, and serum AFP level. Similarly, circulating miR-675 was a significantly higher positive predictor than H19 to predict the cyclin D1 and VEGF protein overexpression while tissue lncRNA-H19 was a significantly higher positive predictor of EGFR. The path diagram also shows that miR-675 was a significant positive predictor of tissue lncRNA-H19 (Figure 3).

### 3.5. ROC Curve Analysis of lncRNA-H19 and miR-675

ROC curves analysis for circulating miR-675 and tissue lncRNA-H19 (tumorous and non-tumorous) were performed and revealed that the sensitivity of miR-675 at cutoff ≥1.15 was significantly higher than tissue lncRNA-H19 at cutoff ≥1.37 for discriminating HCC from controls (95%, 89% respectively). In comparison, the sensitivity of lncRNA-H19 for discriminating the tumorous from the non-tumorous liver tissue samples was 76% at cutoff 10.2 (Table 4) (Figure 4A–C).

### 3.6. Overall Survival in HCC Patients

Overall survival (OS) was illustrated in (Figure 5). The high expressions of tissue lncRNA-H19 and miR675 were associated with low OS (log rank = 34.28, 22.08, respectively, *p* < 0.001).

### 3.7. Cox Regression Analysis for Factors Associated with Low Survival among the Studied Patient Group

Cox regression analyses for predicting patients’ decease were performed and revealed that the most significant predictors of death were high tissue lncRNA-H19, miR-675, multiple focality, LV1 (*p* < 0.001, 0.001, 0.003, and 0.004, respectively) (Table 5).

## 4. Discussion

Elevated expression levels of lncRNA-H19 and miR675 have been found in several malignancies, including inflammatory-mediated HCC, but the roles of these molecules in this disease are debatable. Some studies have suggested that these molecules have tumor suppressor activity, while others have found evidence of pro-tumorigenic activity [11,12,13,14]. The effect of lncRNA-H19 on enhancing the epithelial-mesenchymal transition (EMT) process and metastasis may represent its most significant role in tumorigenesis. H19/miR-675 expression has been demonstrated to be induced by several EMT modulators. As a result, it has been noted that EMT inducers share the H19/miR-675 axis [33].

The finding that the knockdown of lncRNA-H19 prevented the development of HCC further supported the link between lncRNA-H19 and the development of HCC [14,34]. Furthermore, Gamaev et al. [12] demonstrated that in the Mdr2-KO mouse model, the lncRNA-H19 acts as a pro-oncogene throughout the progression of chronic inflammation-induced HCC, primarily by escalating liver damage and lowering hepatocyte polyploidy in young mice.

In the present study, tissue lncRNA-H19 expression was higher in tumorous samples than in non-tumorous and control samples. This could result from a mechanism where Ras GTPase-activating protein-binding protein 1 (G3BP1), a well-known oncoprotein, can specifically bind to m5C-modified H19 lncRNA, causing MYC accumulation to exert its oncogenic effect, closely linked to poor differentiation of HCC [35]. Accordingly, Yang et al. [36] demonstrated that lncRNA-H19 expression was found to be correlated with HCC aggressiveness and poor disease outcomes. In contrast, according to some studies, lncRNA-H19 appears significantly down-regulated in HCC, correlated with poor prognosis [37,38]. Harari-Steinfeld et al. [13] stated that miR-675 is up-regulated in pre-tumorigenic livers along with an increase in lncRNA-H19, which may promote the development of HCC in hepatic cirrhosis and fibrosis. Consequently, miR-675 may be a helpful target therapy for hepatocarcinogenesis [13].

Interestingly, Ye et al. [37] showed that lncRNA-H19 is highly expressed by tumor-associated macrophages, which encourages HCC aggressiveness. Raveh et al. [14] showed that by down-regulating the Fas-associated protein with death domain (FADD), lncRNA-H19 and miR-675 increase gastric cancer, promoting cell proliferation and preventing apoptosis. Furthermore, miR-675, which affects cell cycle regulation by inhibiting retinoblastoma (RB) protein, could control the promotion of HCC tumorigenesis by directly increasing HCC proliferation [13].

Additionally, miR-675 increases the expression of lncRNA-H19, which is crucial for the transition of HCC from apoptosis to necroptosis; further progression of necroptosis intensifies the liver parenchymal inflammatory process [13]. On a molecular level, miR-675 suppresses histone protein 1 (HP1) by reducing tri-methylation of both histone H3 and histone H3 on lysine 9 (H3K9) and 27 (H3K27) and increasing the acetylation of H3K27 at the promotor area of early growth response 1 (EGR1) gene; this increases the transcription of early EGR-1 zinc finger protein, promotes upregulation of lncRNA- H19, and activates tumor-specific pyruvate kinase M2 (PKM2), a process which is necessary for the Warburg effect and gene expression during tumorigenesis as well. Eventually, this leads to the development of tumors and the encouragement of angiogenesis [38].

An intriguing finding in the path diagram analysis was that miR-675, which lncRNA-H19 encodes in its first exon, was a significant positive predictor of lncRNA-H19. As a result, H19 can function as both a reservoir and a miRNA sponge [14,39]. lncRNA-H19 and miR-675 expression levels have been shown to positively correlate in several cancerous tissues, including colorectal cancer, glioma, and gastric cancer [40,41].

Although the expression of miR-675 was more predictive of tumor size and grade than lncRNA -H19, the current study found that lncRNA-H19 and miR-675 were significantly associated with tumor size and grade. This is because miR-675, derived from lncRNA-H19, acts through various mechanisms, including interactions with proteins and/or miRNAs, to maintain the hallmarks of cancer [42]. In addition, the existence of H19 in exosomes involved in tumor progression promotes its significance in this pathology because it is supposed to be a predictor marker for breast, stomach, and lung malignancies and a prognostic marker [43].

Using ROC analysis to evaluate the diagnostic performance of the investigated parameters, the expression level of miR-675 and tissue expression level of lncRNA-H19 had nearly identical Area Under the ROC Curves (AUCs) (0.96) for diagnosing HCC. However, that miR-675 had higher sensitivity at the cutoff point of 1.15 (95% versus 89%). As a non-invasive diagnostic tool for HCC detection, miRNA-675 is therefore preferred. Furthermore, several researchers have assessed the diagnostic usefulness of lncRNA-H19 in distinguishing between cancer and non-cancerous conditions. For example, Yörüker et al. [44] reported that lncRNA-H19 levels were higher in gastric cancer patients compared to cancer-free individuals, and lncRNA-H19 levels significantly decreased after tumors were surgically removed. Additionally, Zhou et al. [45] evaluated the diagnostic utility of lncRNA -H19 level in gastric cancer and reported that with a diagnostic power of 0.838, lncRNA-H19 levels could differentiate between the patients with gastric carcinoma and the healthy controls. Regarding this, the current study found that, at a cutoff value of 10.2 and an AUC of 0.78, lncRNA-H19 expression levels showed moderate sensitivity and specificity for differentiating between tumorous and non-tumorous liver tissue samples (76 and 74 percent, respectively), which was consistent with Fawzy et al. [46], who reported that tissue lncRNA-H19 could distinguish between HCC and chronic liver disease with sensitivity and specificity of (56 percent and 90 percent, respectively) at cut off 19.6 and AUC 0.75. Accordingly, Chen et al. [47] reported that lncRNA-H19 had a diagnostic power of 0.69 in differentiating between tumorous and nearby non-tumorous tissues.

The present study revealed that the rising of lncRNA-H19 and mir-675 levels were significantly related to rising AFP levels. The correlation between AFP levels and the expression levels of lncRNA-H19 was also found by Rojas et al. [1] to be positively correlated. However, they found no association between lncRNA-H19 expression and vascular invasion, portal thrombosis, or the number or location of nodules. Additionally, Hernandez et al. [11] showed that HCCs that secrete AFP are frequently linked to poor prognosis and exhibited increased expression of lncRNA-H19 and its product miR-675, which led to reduced expression of the tumor suppressor RB, which in turn encouraged HCC cell lines to proliferate.

In the current study, high expressions of lncRNA-H19 and Mir-675 were associated with low OS (log rank = 34.28 and 22.08, respectively, *p* < 0.001). The pre-carcinogenic implication of lncRNA-H19 in HCC development is demonstrated primarily by their essential role in establishing a pro-tumorigenic microenvironment, reduction of hepatic cells polyploidy in young animals, and enhancement of proliferation of hepatic cells in all ages [12]. This finding is in line with Rojas et al. [1], who found that patients who died during the follow-up had higher expression of lncRNA -H19 correlated with OS. The level of lncRNA-H19 expression in tumors was also reported by Gamaev et al. [12], associated with an adverse effect on patient survival in HCC. Furthermore, Yang et al. [36] showed that patients’ shorter disease-free survival was related to high lncRNA-H19 levels in resected HCC tissues. Contrarily, Zhang et al. [48] demonstrated that the intra-tumorous tissues had lower levels of lncRNA-H19 expression than the peri-tumorous tissues, and the reduced ratio between them was a standalone predictor of poor outcome in HCC patients.

Angiogenesis is a process related to the progression of tumors from benign to malignant. The high expression of lncRNA-H19 affects tumorigenesis by increasing the expression of several transcripts, including VEGF, which is known to stimulate angiogenesis in endothelial cells [49]. In addition, the knockdown of lncRNA-H19 in Hep3B cells following induction of hypoxic stress leads to alteration of the expression of specific genes related to tumorigenesis, survival, and angiogenesis [15].

Accordingly, the current study demonstrated that increasing levels of lncRNA-H19 and miR-675 were significantly associated with rising levels of EGFR, cyclin D1, and VEGF and that lncRNA-H19 was a significantly higher positive predictor of EGFR, which is a transmembrane glycoprotein from tyrosine kinase receptor (TKR) family that is commonly expressed in epithelial tumors [50].

According to Ito et al. [51], EGFR was expressed in 68% of HCC, suggesting its role in HCC. Furthermore, this suggests that EGFR plays a crucial part in the progression of HCC, as its expression is correlated with a high rate of proliferation, an advanced tumor stage, intrahepatic metastasis, and poor disease-free survival [51]. In contrast, In HCC patients, EGFR is frequently downregulated, and the concurrent upregulation of transforming growth factor (TGF) has prognostic significance [52]. Thankfully, several EGFR inhibitors are being used in clinical settings. For example, Nimotuzumab, a humanized anti-EGFR monoclonal antibody (mAb) that prevents cancerous cells from proliferating, invading, and metastasizing, was first used to treat an 87-year-old HCC patient. Such treatment resulted in complete disease remission, indicating that this mAb may be a promising alternate for HCC therapy, particularly for surgery or chemotherapeutics’ non-responding or poor-responding patients [53].

The present study showed that Mir-675 was a significantly higher positive predictor of VEGF than lncRNA-H19 using linear regression analyses and path diagrams. High levels of VEGF expression have been connected to vascular invasion, portal vein emboli, recurrence of the disease, poor disease-free survival, and OS in HCC patients [54,55]. The main focus of anti-angiogenic cancer therapies for the past two decades was on the influential proangiogenic factors and their receptors (VEGFRs). The VEGF/VEGFR system, one of the major signal transduction pathways, plays a significant role in the pathogenic mechanisms for HCC development. Thus, several molecularly targeted drugs have been developed to counteract angiogenesis by inhibitory molecules controlling the VEGF pathway. For instance, it has been shown that Sorafenib, the first oral multi-kinase inhibitor drug, increases patient survival by reducing tumor angiogenesis and proliferation [56]. In addition to its primary mechanism for inhibiting tumor growth, Sorafenib also exhibits antiangiogenic activity [57]. It was made clear that lncRNA-H19 is essential for angiogenesis [11]; suppressing lncRNA-H19 may also aid in treating HCC [58]. Mir-675 was a much stronger positive predictor of cyclin D1 than lncRNA-H19. Cyclin D1 is a member of the CAK trimeric complex that is required to control the cell cycle and proliferation. Cyclin D1 plays a role in the development and progression of numerous malignancies, including breast, esophageal, bladder, and lung cancers [59,60].

By promoting transcription of survival-related and chemo-resistance-related genes while suppressing transcription of genes related to apoptotic-signaling, lncRNA-H19 is supposed to play a role in the chemotherapeutic resistance in HCC patients as well [14]. Furthermore, according to Ding et al. [61], silencing lncRNA-H19 decreased the expression of the multidrug resistance 1 (MDR1) chemoresistance gene by obstructing mitogen-activated protein kinase/extracellular signal-regulated kinases (MAPK/ERK) signaling [61]. Furthermore, according to Yang et al. [36], knocking down lncRNA-H19 reduced the expression of miR-675, which improved Sorafenib sensitivity by preventing EMT in HCC cells. These findings demonstrated that lncRNA-H19 operated as an oncogene in HCC [62].

HCC has been treated with various angiogenic factors and their receptors up to this point [63]. These results are consistent with the hypothesis that miR-675, derived from Oncofetal lncRNA-H19, reflects tumor dynamics more accurately than lncRNA-H19 and may therefore represent a new biomarker for the diagnosis, prognosis, and monitoring of therapeutic responses in HCC. To clarify the association of the expression of lncRNA-H19 and HCC recurrence and the function of lncRNA-H19 and miR-675 in the molecular pathogenesis of HCC and their potential use as therapeutic targets, experimental studies might be needed in addition to larger cohort studies for an extended period to confirm these findings

## 5. Conclusions

Oncofetal lncRNA-H19-derived mi-R675could be considered a non-invasive biomarker in place of tissue lncRNA-H19 expression as a potential diagnostic and prognostic marker for HCC as well as a candidate for the development of promising therapeutic targets for HCC.

## Figures and Tables

**Figure 1 biomolecules-13-00003-f001:**
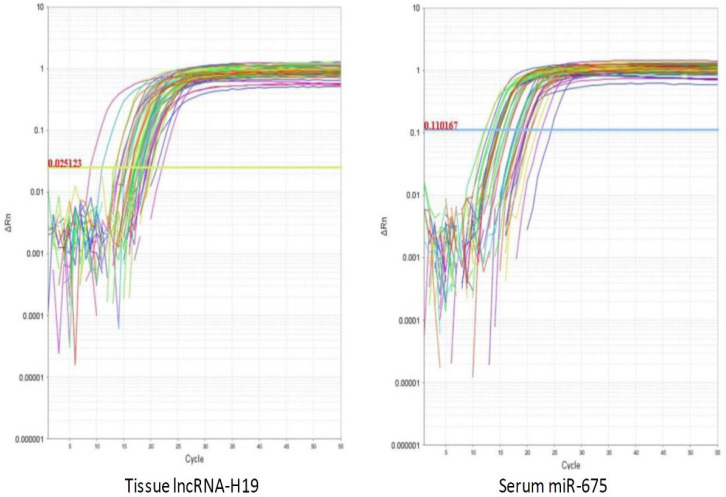
The amplification plots of lncRNA-H19 and miR-675.

**Figure 2 biomolecules-13-00003-f002:**
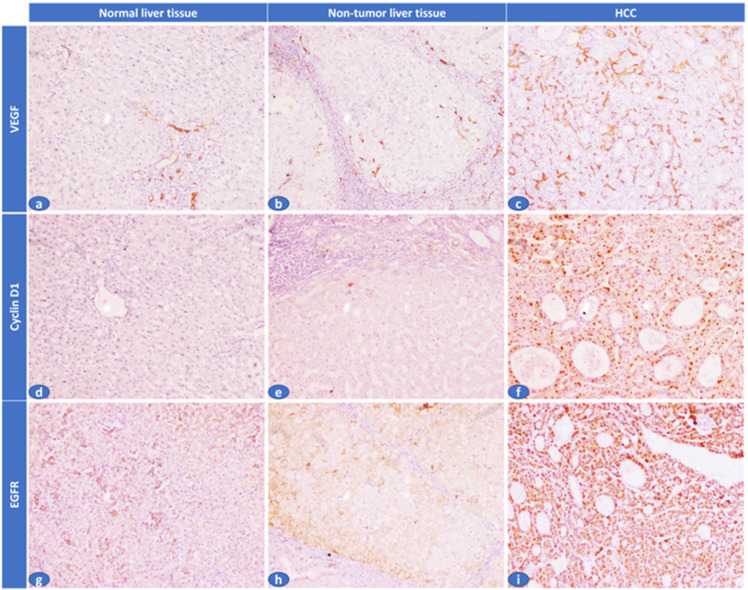
Immunohistochemical expression of VEGF, cyclin D1, and EGFR in the studied cases. (**a**) A normal liver tissue showed absent sinusoidal expression of VEGF (H. score: 0/300), (**b**) Adjacent non-tumor liver tissue showed focal sinusoids expression of VEGF (H. score: 30/300), (**c**) HCC cases showed diffuse sinusoidal expression of VEGF (H. score: 240/300), (**d**) A normal liver tissue showed absent cyclin D1 hepatocyte expression (H. score: 0/300), (**e**) Adjacent non-tumor liver tissue showed mild/focal cyclin D1 hepatocyte expression (H. score: 10/300), (**f**) HCC case showed diffuse/strong cyclin D1 hepatocyte expression (H. score: 255/300), (**g**) A normal liver tissue showed faint EGFR hepatocyte expression (H. score: 20/300), (**h**) Adjacent non-tumor liver tissue showed mild/focal EGFR hepatocyte expression (H. score: 25/300), (**i**) HCC case showed diffuse/strong EGFR hepatocyte expression (H. score: 300/300) (IHC × 100).

**Figure 3 biomolecules-13-00003-f003:**
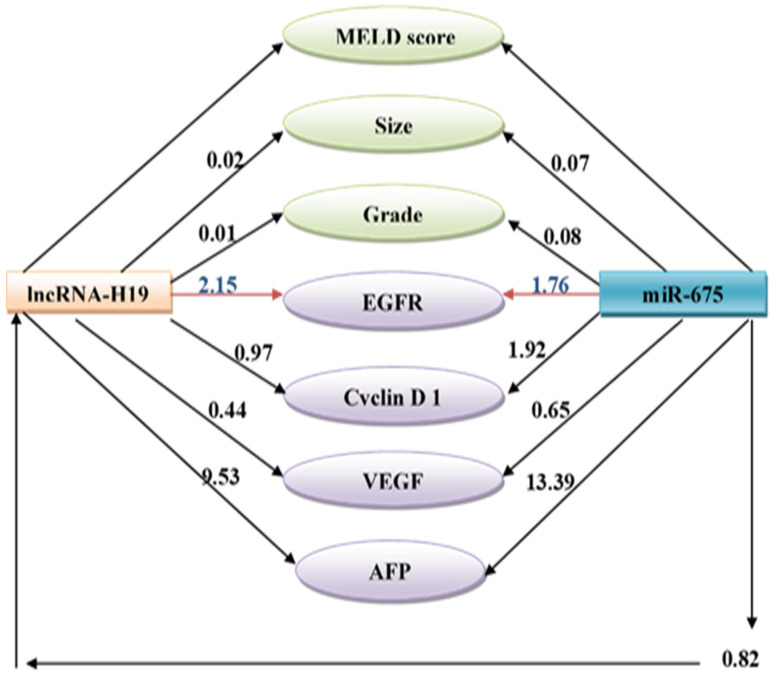
Path analysis diagram of the model used for tumorous samples.

**Figure 4 biomolecules-13-00003-f004:**
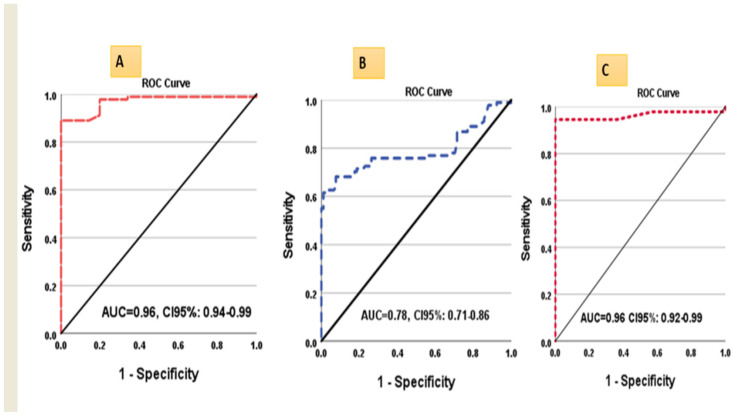
ROC curves for (**A**) lncRNA-H19 in tumorous liver samples vs. controls, (**B**) lncRNA-H19 in tumorous liver samples vs. non-tumorous, and (**C**) miR675 in HCC vs. controls.

**Figure 5 biomolecules-13-00003-f005:**
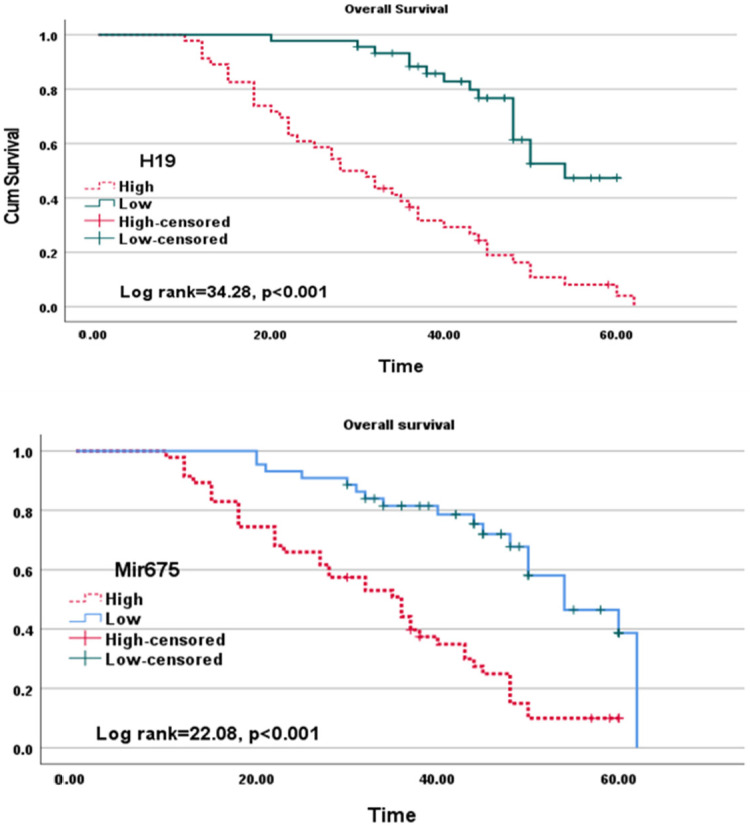
Kaplan–Meier curves of lncRNA-H19 (**upper diagram**) and miR-675 (**lower diagram**) for overall survival.

**Table 1 biomolecules-13-00003-t001:** Clinical characteristics of the studied patients.

Variables		Patients No. = 91	
		No	%
Age at diagnosis(years)	(Mean ± SD)/Range	58.62 ± 5.84/42–67	
Sex:	Male Female	69 22	75.8 24.2
Hepatitis	HCV Non-HCV	83 8	91.2 8.8
Previous HCV treatment	DAAs Interferon Non	50 13 20	60.24 15.66 24.09
Portal hypertension	Yes No	64 27	70.3 29.7
Focal lesions	Single Multiple	79 12	86.8 13.2
Pathological grade	Well differentiated Moderately differentiated Poorly differentiated	15 34 42	16.5 37.4 46.2
Adjacent non-tumor liver fibrosis	Cirrhosis Non-cirrhosis	60 31	65.9 34.1
Adjacent non-tumor liver activity	Mild Moderate	51 40	56.0 44.0
Pathological stage	I, II III, IV	43 48	47.3 52.7
MELD	(Mean ± SD)/Range	7.36 ± 1.33	6–11
Size	(Mean ± SD)/Range	6.92 ± 3.11	2.5–15.5
Survival status	Dead Alive	58 33	63.7 36.3

DAAs: Direct acting non-viral agents.

**Table 2 biomolecules-13-00003-t002:** Biochemical and histopathological markers among the studied groups.

	Controls (No. = 91)	HCC Cases		*p* Value	Effect Size (95%CI)
		Non-Tumorous (No. = 91)	Tumorous (No. = 91)		
	Median (IQR)	Median (IQR)	Median (IQR)		
EGFR	100(90–120)	120(90–200)	180(120–235)	<0.001 *	0.31[0.23–0.37]
Cyclin D1	15(10–50)	50(20–100)	120(60–150)	<0.001 *	0.48[0.42–0.54]
VEGF	5(2–5)	10(5–10)	65(45–80)	<0.001 *	0.76[0.73–0.79]
lncRNA-H19	0.9(0.8–1.2)	4.5(1.9–10.9)	19.5(10.3–33.5)	<0.001 *	0.65[0.60–0.69]
miR-675	1(0.9–1.1)		11.6(1.5–18.3)	<0.001 *	-
AFP	15(10–50)		120(60–150)	<0.001 *	-

*: significant.

**Table 3 biomolecules-13-00003-t003:** Correlation between lncRNA-H19 and miR-675 with demographic and biochemical markers among the HCC patients.

	LncRNA-H19			miR675		
	rs	(95% CI)	*p* Value	rs	(95% CI)	*p* Value
Age	0.30	0.09–0.48	0.004 *	0.41	0.22–0.57	<0.001 *
MELD score	0.13	−0.07–0.33	0.198	0.08	−0.13–0.29	0.426
Size	0.53	0.36–0.67	<0.001 *	0.42	0.23–0.85	<0.001 *
Pathological grade	0.43	0.24–0.59	<0.001 *	0.41	0.22–0.57	<0.001 *
EGFR	0.71	0.19–0.55	<0.001 *	0.39	0.19–0.55	<0.001 *
Cyclin D1	0.39	0.19–0.55	<0.001 *	0.32	0.12–0.50	0.002 *
VEGF	0.45	0.26–0.60	<0.001 *	0.31	0.10–0.49	0.002 *
AFP	0.71	0.58–0.80	<0.001 *	0.64	0.49–0.75	<0.001 *
miR-675	0.45	0.26–0.60	<0.001 *	-	-	-

*: significant.

**Table 4 biomolecules-13-00003-t004:** ROC curve analysis of lncRNA-H19 for discrimination between tumorous samples and non-tumorous samples, miR-675 for discrimination between HCC and controls.

	Tumorous Samples vs. Controls		Tumorous vs. Nontumorous Sample
	LncRNA-H19	miR-675	LncRNA-H19
AUC [95% CI]	0.96 [0.94–0.99]	0.96 [0.92–0.99]	0.78 [0.71–0.86]
Cutoff point	≥1.37	≥1.15	≥10.20
Sensitivity % [CI95%]	89 [80–94]	95 [87–98]	76 [66–84]
Specificity % [CI95%]	86 [76–92]	82 [73–89]	74 [63–82]
Accuracy % [CI95%]	87 [81–92]	88 [83–93]	75 [68–81]
PPV% [CI95%]	86 [77–92]	84 [75–90]	74 [64–82]
NPV% [CI95%]	89 [80–94]	94 [85–98]	75 [65–84]

**Table 5 biomolecules-13-00003-t005:** Cox regression analysis for factors associated with low survival among the studied patient group.

	*p* Value	Hazard Ratio	(95% CI)	
			Lower	Upper
LncRNA-H19	<0.001 *	1.03	1.01	1.04
miR-675	0.001 *	1.04	1.02	1.06
Focality (Multiple)	0.003 *	2.90	1.46	5.79
LV1 (Yes)	0.004 *	5.36	1.73	16.57
Adjacent liver (non-cirrhosis)	0.154	0.62	0.32	1.19
Size (≥5)	0.354	0.64	0.25	1.63
Pathological grade (poorly differentiated)	0.501	0.81	0.43	1.51
Pathological stage (III + IV)	0.760	0.82	0.22	2.99

Significant: *, (LVI) Lymph vascular invasion.

## Data Availability

The data presented in this study are available on request from the corresponding author.

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
