# Peer review of "The Potential Utility of Circulating Oncofetal H19 Derived miR-675 Expression versus Tissue lncRNA-H19 Expression in Diagnosis and Prognosis of HCC in Egyptian Patients"

_biomolecules, 2022, doi:10.3390/biom13010003_

Round 1
Reviewer 1 Report
The article by Abdelsattar describes a study to demonstrate the diagnostic significance of circulating miR-675 over tissue-derived lncRNA-H19. The authors used qPCR and IHC analysis in HCC patients and Healthy individuals to prove their hypothesis. Although the hypothesis is interesting but is not novel and both of these miR and lncRNA have been implicated in different cancers. The article also needs to be restructured again and needs a major revision before it is accepted for publication.
Please see following suggestions
1. The correlation between miR675, lncRNA-H19 and EGFR, VEGF is based on the expression profile but no experimental evidence is given to show the molecular interactions between them. This is a major concern
2. More experiments like co-IP, luciferase assays etc is needed to implicate the interactions and also authors should give a putative description of the signaling pathway involved in the process.
3. Authors should perform a miR mimic or miR anatoginst study to show the effect in cancer models.
4. the method section needs to be rewritten
5. The staining figures (IHC) are very confusing and should be labeled properly.
6. The results should be discussed under different subheadings, it is very difficult to follow the results and correlate them with figures and tables and understand the outcome of the study
Reviewer 2 Report
In the current manuscript, the author focused on comparing the expression levels of long noncoding RNA H19 and miR675 between HCC patients and healthy controls. The author pointed out that H19 and miR675 have the potential to serve as diagnostic markers for HCC and the expression level of H19 and miR675 correlate with HCC progression. The methods used in the manuscript are rational and the results are supportive of the conclusion.
However, I have one major question or concern about this study regarding the samples sued to quantify H19 and miR675. In the current study, the author measured serum miR675 and tissue H19. While the tissue H19 level may represent the actual expression of H19 in liver tissues, the circulating miR675 level may be affected by many other factors. Comparing the diagnostic sensitivity or accuracy regarding the expression level of H19 and miR765 from different types of samples is not very appropriate nor convincing. The absolute amount of RNA amount in serum and tissues have great differences, also the detection of serum miRNA expression suffers from high background “noise” from other tissues (serum miR675 are not generated from the liver alone). A good explanation about why choosing serum miR675 and tissue H19 here is needed. As the author has collected both serum and liver tissue samples, it would be great if the author can measure H19 and miR675 in both samples and perform the analysis and comparison.
Several other suggestions were listed below to help improve the manuscript's quality.
1. In the method section line 121, does the author also collect tissues from organs other than the liver? If not, I would suggest replacing the statement “tissue samples” with a more specific statement “liver tissue samples” across the manuscript. Also, I would suggest mentioning the clinically used diagnostic method to distinguish the tumorous and non-tumorous liver tissues here.
2. I suggest the author adding the qPCR result figures to the manuscript.
3. As the author already quantified the IHC results (line 192-193), I would suggest adding these results to corresponding figures (Fig1).
4. Grammar issues were found in several places in the current manuscript. I would suggest the author carefully proofread the content and correct these issues. Examples: incomplete sentence at line 104-105; missing period symbol at line 48 and 114; improper capitalized word at line 312.
5. The use of abbreviations needs to be revised. Example: full name of event per variable (EPV) appeared after the use of its abbreviation at line 196; the full name of CAK appeared twice in the manuscript.
6. In several places across the manuscript, the author needs to revise the references. Line 65-66, a reference is needed here to support the statement. Line 88, Ref 21 is not about H19 here.
Round 2
Reviewer 1 Report
The authors have adequately answered my concern and this manuscript is in an acceptable format